

**Modelling tree-ring cellulose δ[18]O variations of two temperature-sensitive tree**
**species from North and South America**
**Authors:**
Aliénor Lavergne[1], Fabio Gennaretti[1], Camille Risi[2], Valérie Daux[3], Etienne Boucher[4], Martine
M. Savard[5], Maud Naulier[6], Ricardo Villalba[7], Christian Bégin[5] and Joël Guiot[1]
[1]Aix Marseille Université, CNRS, IRD, Collège de France, CEREGE, ECCOREV, Aix-en-
Provence, France
[2]Laboratoirede Météorologie Dynamique, IPSL, UPMC, CNRS, Paris, France
[3]Laboratoire des Sciences du Climat et de l'Environnement, CEA-CNRS-UVSQ, 91191 Gif-sur-
Yvette, France
[4]Department of Geography and GEOTOP, Université du Québec à Montréal, Montréal, Canada
[5]Geological Survey of Canada, Natural Resources Canada, 490 rue de la Couronne, QC,
G1K9A9, Canada
[6]Institut de Radioprotection et de Sureté Nucléaire (IRSN), PRP-ENV, SERIS/LRTE, Saint-Paul-
lez-Durance, France
[7]Instituto Argentino de Nivología, Glaciología y Ciencias Ambientales, IANIGLA-CONICET,
Mendoza, Argentina
**Corresponding authors:** Aliénor Lavergne (alienor.lavergne@gmail.com) and Fabio Gennaretti
(gennaretti@cerege.fr)
Tel : +33 (0) 4 42 97 15 32
Centre Européen de Recherche et d'Enseignement en Géosciences
Technopôle de l'Arbois-Méditerranée
13545 Aix-en-Provence, FRANCE





**ABSTRACT**
Oxygen isotopes in tree-rings ($\delta^{18}O_{TR}$) are widely used to reconstruct past climates. However, the
complexity of climatic and biological processes controlling isotopic fractionation is not yet fully
understood. Here, we use the MAIDENiso model to decipher the variability of $\delta^{18}O_{TR}$ of two
temperature-sensitive species of relevant paleoclimatological interest (*Picea mariana* and
*Nothofagus pumilio*) and growing at cold high-latitudes in North and South America. In this first
modelling study on $\delta^{18}O_{TR}$ values in both northeastern Canada (53.86°N) and western Argentina
(41.10°S), we specifically aim at: 1) evaluating the predictive skill of MAIDENiso to simulate
$\delta^{18}O_{TR}$ values, 2) identifying the physical processes controlling $\delta^{18}O_{TR}$ by mechanistic modelling
and, 3) defining the origin of the temperature signal recorded in the two species. Although the
linear regression models used here to predict daily $\delta^{18}O$ of precipitation ($\delta^{18}O_P$) may need to be
improved in the future, the resulting daily $\delta^{18}O_P$ values adequately reproduce observed (from
weather stations) and simulated (by global circulation model) $\delta^{18}O_P$ series. The $\delta^{18}O_{TR}$ values of
the two species are correctly simulated using the $\delta^{18}O_P$ estimation as MAIDENiso input, although
some offset in mean $\delta^{18}O_{TR}$ levels is observed for the South American site. For both species, the
variability of $\delta^{18}O_{TR}$ series is more likely linked to the effect of temperature on isotopic
enrichment of the leaf water rather than on the isotopic composition of the source water. We
show that MAIDENiso is a powerful tool for investigating isotopic fractionation processes but
that the lack of a denser isotope-enabled monitoring network recording oxygen fractionation in
the soil-vegetation-atmosphere compartments limits our capacity to decipher the processes at
play. This study proves that the eco-physiological modelling of $\delta^{18}O_{TR}$ values is necessary to
interpret the recorded climate signal more reliably.

**Keywords:** MAIDENiso model, $\delta^{18}O$, tree-ring, *Nothofagus pumilio*, *Picea mariana*





## 1. INTRODUCTION


Oxygen isotopes in tree rings ($\delta^{18}O_{TR}$) are increasingly used as indicators of past climatic
changes in temperate areas (Cernusak and English, 2015; Hartl-Meier et al., 2014; Saurer et al.,
2008). They have been widely used to reconstruct past atmospheric conditions such as air
temperature (Naulier et al., 2015), drought (Labuhn et al., 2016), precipitation amount (Rinne et
al., 2013), isotopic composition of precipitation (Danis et al., 2006), relative air humidity
(Wernicke et al., 2015), cloud cover (Shi et al., 2012), and even atmospheric circulation patterns
(Brienen et al., 2012). This diversity of climatic targets possibly reconstructed based on oxygen
isotopes hints at the challenge of understanding the complexity of the climatic and biological
processes that control isotopic fractionation of oxygen in trees (Treydte et al., 2014).
Uncertainties arise because different poorly measured factors influence $\delta^{18}O_{TR}$ values. Isotopic
signals in tree-rings cellulose are strongly influenced by isotopic signature of soil water taken up
by the roots and by evaporative and physiological processes occurring at the leaf level and during
downstream metabolism (Barbour et al., 2005; Gessler et al., 2014). Thus, a comprehensive
approach that embraces existing mechanistic understanding of the fractionation processes
involved is required.

Few isotopic process-based models have been developed to investigate the mechanistic rules
governing the $\delta^{18}O_{TR}$ variations (Guiot et al., 2014): the Péclet-modified Craig-Gordon model
(Kahmen et al., 2011) and the Roden's model (Roden et al., 2000) are able to estimate, at a daily
time step, the $\delta^{18}O$ values of soil and xylem waters, and the isotopic fractionation occurring in the
leaves due to evapotranspiration. Versions of these models are integrated in more complete forest
ecophysiological models simulating the ensemble of forest water and carbon fluxes: (1)
MAIDEN (Modeling and Analysis In DENdroecology) (Gea-Izquierdo et al., 2015; Misson,
2004), which contains the isotopic module MAIDENiso (Danis et al., 2012) and (2) MUSICA
(Ogée et al., 2003, 2009). Both are accounting for important post-photosynthetic factors and are
able to link photosynthesis and carbohydrate allocation to stem growth.

In this paper, we use the MAIDENiso model to decipher the $\delta^{18}O_{TR}$ variability in American
temperature-sensitive species (*Picea mariana* in northeastern Canada and *Nothofagus pumilio* in
western Argentina). The selected sites are of special interest for paleoclimatology given that their





$\delta^{18}O_{TR}$ chronologies carry strong temperature signals. A summer temperature reconstruction was
already developed at the North American site (Gennaretti et al., 2017a; Naulier et al., 2015) and a
calibration study conducted at the South American one highlighted the strong potential of $\delta^{18}O_{TR}$
values to reflect variations in summer-autumn temperatures over a large region south of 38°S
(Lavergne et al., 2016). However, up to now, the climate-$\delta^{18}O_{TR}$ relationships were analysed
using a black box approach based on linear models. Here, we specifically aim at: 1) evaluating
the predictive skill of MAIDENiso to simulate $\delta^{18}O_{TR}$ values, 2) identifying the physical
processes controlling $\delta^{18}O_{TR}$ by mechanistic modelling and, 3) defining the origin of the
temperature signal recorded in the two species.

**2.   DATA AND METHODS**

**2.1. Sampling sites and tree-ring data**

Two high-latitude American native species were studied here: 1) *Picea mariana* (Mill. B.S.P.;
black spruce), which is a conifer widely distributed over the American boreal forest (Viereck and
Johnston, 1990); and 2) *Nothofagus pumilio* (Poepp. et Endl. Krasser; lenga), which is a
deciduous species dominating the high-elevation forests along the Patagonian Andes from 35°S
to 55°S (Donoso, 1981; Schlatter, 1994). We selected two sites of *P. mariana* in the centre of the
Quebec-Labrador Peninsula in northeastern Canada (L01 and L20; from 53°51'N-72°24'W to
54°33'N-71°14'W, ~480 m elevation; see Gennaretti et al. (2014) and Naulier et al. (2014) for
details) and three sites of *N. pumilio* in northern Patagonia, western Argentina (NUB, ALM and
CHA; from 41°09′S-71°48′W to 41°15′S-71°17′W, 1270-1610 m elevation; see Lavergne et al.
(2016, 2017) for details). Climate in northeastern Canada is mostly continental and subarctic with
short, mild and wet summers and long, cold and dry winter. Total annual precipitation averages
825 mm with up to 46% falling during the growing season in summer (June to September)
(Naulier et al., 2014). In western Argentina, precipitation is largely concentrated from late fall to
early spring followed by a drier and mild period during summer and early fall (López Bernal et
al., 2012).

Four trees per site were collected for both species. The selection of the samples and analytical
procedure for $\delta^{18}O_{TR}$ measurements were described in Lavergne et al. (2016) and Naulier et al.





(2014). The developed $\delta^{18}O_{TR}$ chronologies covered the 1950-2005 and 1952-2011 periods at the
northeastern Canadian and western Argentinian sites, respectively. For each species, the
chronologies obtained at the different stands being significantly inter-correlated (Figure 1), we
chose to combine them and to develop one isotopic chronology for each of the two species.

**2.2. Modelling oxygen isotopes in tree-ring cellulose with MAIDENiso**
MAIDENiso is a process-based model that can simulate in parallel phenological and
meteorological controls on photosynthetic activity and carbon allocation. It explicitly allocates
carbohydrates to different carbon pools (leaves, stem, storage and roots) on a daily basis using
phenological stage-dependent rules. It also simulates the fractionation of carbon and oxygen
isotopes during growth processes. In particular, it estimates at a daily time step $\delta^{18}O$ values of
soil water and xylem water, the isotopic fractionation occurring in the leaves due to
evapotranspiration and the biochemical fractionation during cellulose formation. It uses as input
daily maximum and minimum temperature (°C), precipitation (cm/day), atmospheric $CO_2$
concentration (ppm) and $\delta^{18}O$ values of precipitation ($\delta^{18}O_P$ in ‰).

In this study, the calculation of the daily $\delta^{18}O_{TR}$ in tree-ring cellulose (‰) is based on the (Danis
et al., 2012)'s formulation of the Craig-Gordon model (Craig and Gordon, 1965):
$$\delta^{18}O_{TR} = (1-f_o)\cdot[\varepsilon^* +\varepsilon_k \cdot(1-h_{air}) + h_{air}\cdot\delta^{18}O_V + (1-h_{air})\cdot\delta^{18}O_{XW}]+f_o\cdot \delta^{18}O_{XW} +\varepsilon_0 \quad (1)$$
This equation summarizes how $\delta^{18}O_{TR}$ is determined by:
(i)   the $\delta^{18}O$ of the source (xylem) water ($\delta^{18}O_{XW}$), which is computed by averaging the

$\delta^{18}O_{SW}$ values of the different soil layers weighted by the volume of water taken up by

the roots in each layer. The isotopic effects of water mixing and soil evaporation on

the $\delta^{18}O_{SW}$ values of the different soil layers are computed by a mass and isotopic

balance (Danis et al., 2012). It is worth noting that no fractionation occurs during

water uptake by roots (Wershaw et al., 1966), neither during the transport of water

from the roots to the leaves.

(ii)  the $^{18}O$ enrichment of the leaf water due to transpiration is described by

$(\varepsilon^*+\varepsilon_k\cdot(1-h_{air})+h_{air}\cdot\delta^{18}O_V+(1-h_{air})\cdot\delta^{18}O_{XW})$ after (Craig and Gordon, 1965), where:





a. $\varepsilon^*$ is the equilibrium fractionation due to the change of phase from liquid water to

vapour at the leaf temperature (fixed at 21.4°C, the temperature threshold for

maximum carbon assimilation, $\varepsilon^*$ is 9.65‰ (Helliker and Richter, 2008)),

b. $\varepsilon_k$ is the kinetic fractionation due to the diffusion of vapour into unsaturated air

through the stomata and the leaf boundary layer,

c. $h_{air}$ is the relative humidity of the evaporating air mass estimated from daily air

temperature ($T_{air}$; °C; mean of the maximum and minimum air temperatures), and

the dew point temperature ($T_r$; °C) (Running et al., 1987),

154       d. $\delta^{18}O_V$ is the atmospheric water vapour calculated assuming a precipitation-vapour

isotopic equilibrium (see below);

(iii)    the biochemical fractionations ($\varepsilon_0$) due to oxygen exchange between carbonyl groups

(C = O) in the organic molecules and water (DeNiro and Epstein, 1979; Farquhar et

al., 1998).

(iv)    the dampening factor $f_o$ reflecting the exchange of the oxygen atoms between sucrose

and xylem water during cellulose synthesis in the xylem cells of tree rings.

As previously evoked (i), $\delta^{18}O_{XW}$ of Eq. 1 depends on $\delta^{18}O_{SW}$ and thus on $\delta^{18}O_P$ values. However,
long continuous time series of $\delta^{18}O_P$ are not available in the studied area. Here, we tested the
impact of using two different methods for deriving $\delta^{18}O_P$ time series.
Firstly, a linear model was used to estimate the daily values of $\delta^{18}O_P$ and subsequently $\delta^{18}O_V$
based on the primary drivers of their temporal variability (Dansgaard, 1964; Horita and
Wesolowski, 1994), that are air temperature ($T_{air}$; °C) and precipitation at the corresponding site
(P; mm):

$\delta^{18}O_P = a \cdot T_{air} + b \cdot P + c$                         (2)

$\delta^{18}O_V = \delta^{18}O_P - \varepsilon^*_{Tair}$                              (3)

with $\varepsilon^*_{Tair}$ the fractionation due to the change of phase from liquid water to vapour at the mean air
temperature. The coefficients $a$ and $b$ were allowed to vary over a plausible range (or prior range)
in the calibration process together with other MAIDENiso parameters, while coefficient $c$ was
fixed to a likely value (see Table 1 and section 2.4). This estimated set of data is referred in the
following as the estimated $\delta^{18}O_P$ dataset.



Secondly, we run the model with the series of the daily $\delta^{18}O_P$ derived from two general
circulation models (GCM) with different spatial resolutions and enough available data at our site
locations: 1) the MUGCM model (Noone and Simmonds, 2002) forced by varying sea surface
temperature (SST) from the HadISST data set for the 1950-2003 period (2°×2° resolution;
extracted at http://paos.colorado.edu/~dcn/SWING/database.php ; hereafter referred as MUGCM
$\delta^{18}O_P$ dataset), and 2) the LMDZ5A model (Hourdin et al., 2013; Risi et al., 2010) with the
horizontal winds guided by those of the NCEP20 reanalysis for the 1950-2008 period (Compo et
al., 2011) (2.5°×3.75° resolution; hereafter referred as LMDZ-NCEP20 $\delta^{18}O_P$ dataset).

The final $\delta^{18}O_{TR}$ time series are the annual average of the $\delta^{18}O_{TR}$ daily values (Eq. 1) weighted by
the daily simulated stand Gross Primary Production (GPP), assuming a proportional allocation of
carbon to the trunk. For the northeastern Canadian sites, the GPP simulated by MAIDENiso was
optimized using observations from an eddy covariance station (see Gennaretti et al. (2017a)).
Unfortunately, such observations were not available for *N. pumilio*, therefore the
parameterization obtained for the GPP of *P. mariana* was also used for the western Argentinian
sites but constraining the simulations with phenological observations extracted from the
literature. For example, to respect the annual cycle of the leaf area index (LAI) for *N. pumilio*
(Magnin et al., 2014; Rusch, 1993), we used in MAIDENiso a seasonal LAI annual cycle with a
development of leaves (LAI increase) between October and November, a maximum LAI (set at 5
leaf area/ground area) from November to April, a decreasing LAI (leaf fall) between April and
May, and finally a leafless period (null LAI) from June to September (Magnin et al., 2014;
Rusch, 1993). Furthermore, based on the finding that $\delta^{18}O_{TR}$ annual time series were more
correlated to climate variables of specific months of the growing season (Lavergne et al., 2016),
we also computed $\delta^{18}O_{TR}$ annual values by weighting the $\delta^{18}O_{TR}$ daily values (Eq. 1) with
synthetic GPP time series maximizing the correspondence between observations and simulations.

**2.3. Meteorological and atmospheric CO$_2$ data**
At the western Argentinian sites, we did not have long daily records of observed climate data.
Therefore, daily minimum–maximum temperature and precipitation data were derived from the
20th Century Reanalysis V2c (Compo et al., 2011) provided by the NOAA/OAR/ESRL (2°×2°
resolution, https://www.esrl.noaa.gov/psd/data/gridded/data.20thC_ReanV2c.html). The



temperature daily time series of the reanalysis were corrected in order to respect the monthly
mean values detected at Bariloche, the nearest meteorological station from our sampling sites
(~48 km from the sites, 41°12′ S–71°12′ W, 840 m asl; Servicio Meteorológico Nacional,
Argentina). The resulting maximum and minimum temperature series, covering the 1952-2011
period, fit well with the daily local temperature data from La Almohadilla (ALM) site (41°11'S,
71°47'W, 1410 m asl; data measured by dataloggers and provided by IANIGLA) available over
the 2002-2012 period ($r = 0.74$, $p < 0.001$; Figure SM1). For the northeastern Canadian sites,
climate data were obtained from the gridded interpolated Canadian database of daily minimum–
maximum temperature and precipitation covering the 1950-2005 studied period (0.08°×0.08°
resolution, (Hutchinson et al., 2009); http://cfs.nrcan.gc.ca/projects/3/4). In addition to these data
we also used for both the western Argentinian and northeastern Canadian sites modelled daily
data from the GCMs described above (see Table 2 with the input data used for each tested
configuration).

Data on the atmospheric $CO_2$ concentration were derived from the Mauna Loa station over the
1958-2012 period (Keeling et al. (1976); http://www.esrl.noaa.gov/gmd/ccgg/trends/). For the
1950-1957 years, we extrapolated atmospheric $CO_2$ data using the trend and seasonal cycle
observed in the observations over the subsequent 10-years period (1958-1967).
**2.4. Estimation of parameters influencing $\delta^{18}O_{TR}$**
We used a Bayesian method for the simultaneous calibration of the various MAIDENiso
parameters specific to the study species and site. A set of 50 plausible blocks of parameters
(posterior values) was selected according to the method described in Gennaretti et al. (2017a)
using Markov Chain Monte Carlo (MCMC) sampling (Table 1). The following prior plausible
ranges were considered:
1) the prior ranges of the *a* and *b* coefficients in the equation of the daily $\delta^{18}O_P$ (Eq. 2) were
selected in order to get $\delta^{18}O_P$ values for each site consistent with the measured monthly local
values from the nearest stations of the Global Network of Isotopes in Precipitation (GNIP), and
with the simulated daily values from the LMDZ-NCEP20 model and from the MUGCM model
(see Table 1),



2) the range for the biochemical fractionation factor $\varepsilon_0$ was chosen between 24‰ and 30‰
(+27±3‰ after DeNiro and Epstein (1981); Sternberg (1989); Yakir and DeNiro (1990)),
3) the range for the kinetic fractionation $\varepsilon_k$, which has been set to 26.5‰ in Farquhar et al. (1989)
but that can vary over larger ranges (Buhay et al., 1996), was taken between 10‰ and 30‰ here,
4) the range for the dampening factor $f_o$ was allowed to vary between 0.3 and 0.5 following
Saurer et al. (1997).

We tested the sensitivity of the MAIDENiso model to the calibrated parameters by modifying
them within their respective prior calibration range. To control the robustness of the calibrated
parameters, we performed the calibration of these parameters over two equal length intervals
(1950-1977 and 1978-2005 for *P. mariana*; 1952-1981 and 1982-2011 for *N. pumilio*) keeping
the second half for independent validation of the parameters estimates. Once the model was
calibrated for the two species, the MAIDENiso's performance to simulate *P. mariana* and *N.*
*pumilio* $\delta^{18}O_{TR}$ interannual data was evaluated using the correlation coefficients (r) and the root
mean square errors (RMSE) between observed and simulated values.

**2.5. Disentangling leaf-level fractionation processes and source water influences on**

**$\delta^{18}O_{TR}$ signature**

To define the relative contributions to the $\delta^{18}O_{TR}$ signature of the isotopic signal of the source
water (xylem water) and of the fractionation processes due to transpiration taking place in the
leaves, we designed two experimental simulations with MAIDENiso based on Eq. 1:

1) to quantify the influence of the variability of the isotopic composition of the xylem water

on $\delta^{18}O_{TR}$, we compared the reference simulations to those where the relative humidity

($h_{air}$) and the isotopic composition of atmospheric vapour ($\delta^{18}O_V$) were assumed to be

constant. The constant values for $h_{air}$ and $\delta^{18}O_V$ were defined as the averages of the

respective MAIDENiso outputs ($h_{air}$ = 0.62 and 0.9, and, $\delta^{18}O_V$ = -26.28‰ and -17.34‰,

respectively for northeastern Canada and western Argentina; the XW_source experiment

simulation hereafter),

2) to quantify the influence of the isotopic enrichment of the leaf water due to transpiration

on $\delta^{18}O_{TR}$, we compared the reference simulations to those where the $\delta^{18}O_{XW}$ series were

assumed to be constant. The constant value for $\delta^{18}O_{XW}$ was estimated as the average of





the $\delta^{18}O_{XW}$ MAIDENiso outputs ($\delta^{18}O_{XW}$ = -13.81‰ and -7.03‰, respectively for
northeastern Canada and western Argentina; the Leaf_water_enrichment_driven
experiment simulation hereafter).
Comparison between the experimental and reference simulations (i.e. using the optimal values of
the parameters) was achieved through the calculation of the coefficient of determination ($R^2$).

**3.  RESULTS**
**3.1. Estimated versus modelled and observed $\delta^{18}O_P$ values**
The modelled $\delta^{18}O_P$ series from the GCM models are similar to the GNIP datasets, with mean
values ranging from -12‰ to -8‰ over June-September in northeastern Canada (Figure SM2A)
and from -7‰ to -3‰ over December-April at the western Argentinian sites (Figure SM2B). In
general, $\delta^{18}O_P$ series from LMDZ-NCEP20 model in western Argentina are slightly displaced
toward higher values (+1‰) in comparison with the GNIP and MUGCM data. The estimated
$\delta^{18}O_P$ values based on plausible values of coefficients *a* and *b* agree well with those of the models
and observations in northeastern Canada. For the western Argentinian sites, they are 2-3‰ lower
from April to October, i.e. late spring-early autumn (Figure SM2).

**3.2. Sensitivity of the model to the calibrated parameters**
Most of the calibrated parameters have an influence on the correlations between observed and
simulated $\delta^{18}O_{TR}$ series and/or on the mean levels of the simulated series (Figure 2). The
temperature and precipitation dependences of $\delta^{18}O_P$ values (respectively *a* and *b*) have the
strongest influence on correlations. Increasing *a* and *b* values increase the mean $\delta^{18}O_{TR}$ levels,
most strongly in western Argentina than in northeastern Canada (Figure 2). Changes in the
dampening factor ($f_o$) and in the biochemical fractionation ($\varepsilon_0$) have almost no effect on
correlation, but their increase induces significant decrease of the mean levels of $\delta^{18}O_{TR}$ series.
Finally, increasing the kinetic fractionation ($\varepsilon_k$) leads to lower correlations and to higher mean
levels of $\delta^{18}O_{TR}$ (Figure 2).

**3.3. MAIDENiso performance in reproducing observed $\delta^{18}O_{TR}$ series**
Split-period verifications of the calibrated relationships for *P. mariana* and *N. pumilio* when
using estimated $\delta^{18}O_P$ series from Eq. 2 indicate that the calibration over either the first half or the





second half periods provide similar posterior densities of the calibrated parameters than the ones
obtained when calibrating over the whole periods (Figure SM3). One exception is observed in the
calibration of coefficient *a* in northeastern Canada over the two half periods, where the posterior
densities of *a* are different from the one obtained by calibrating over the entire period. Over the
entire periods, observed and simulated $\delta^{18}O_{TR}$ series are significantly correlated in northeastern
Canada (r = 0.56, p < 0.01 and RMSE = 0.67; Figure 3A) and in western Argentina (r = 0.48, p <
0.01 and RMSE = 0.63; Figure 3C). The correlation between observed and simulated $\delta^{18}O_{TR}$
series are slightly improved when we used synthetic daily GPP (r = 0.62 and r = 0.52, p < 0.01,
respectively for northeastern Canada and western Argentina; Figure 3B and 3D). It is worth
noting that the mean levels of the simulated $\delta^{18}O_{TR}$ series for the Argentinian sites are lower than
those of the observations (offset of around -2.5‰; Figure SM4). The series were therefore
corrected to respect the mean values detected in the observations (Figure 3C and 3D). In contrast,
the correlations between observation and simulation considerably decrease when we used
modelled $\delta^{18}O_P$ from MUGCM models or LMDZ-NCEP20 reanalysis data. They only reach
r = 0.13 (p > 0.05) to 0.23 (p < 0.05) in northeastern Canada and r = 0.23 to 0.26 (p < 0.05) in
western Argentina, respectively (Figure 4).

**3.4. Influence of source water and leaf water isotopic enrichment to the $\delta^{18}O_{TR}$ signature**
The relative contributions to the $\delta^{18}O_{TR}$ signature of the isotopic signal of the source (xylem)
water and of the $^{18}O$ enrichment of the leaf water due to transpiration were investigated. In both
regions, the Leaf_water_enrichment experimental simulations are more highly related to the
reference one ($R^2$ centred on 0.9 and 0.95, respectively for northeastern Canada and western
Argentina; Figure 5) than the XW_source ones ($R^2$ centred on 0.65 and 0.8, respectively for
northeastern Canada and western Argentina). This suggests that, with the model, the variability of
$\delta^{18}O_{XW}$ has a weaker influence on $\delta^{18}O_{TR}$ variations than the changes of the leaf water isotopic
enrichment do. Notably, *P. mariana* in northeastern Canada appears to be more sensitive to both
influences than *N. pumilio* in western Argentina (Figure 5). Caution should be made here since
these results are limited upstream by the performance of the regressions.



## 4. DISCUSSION

### 4.1. Precipitation $\delta^{18}O_P$ variations and estimation

Although the regression models used to predict daily $\delta^{18}O_P$ values are likely too simplistic, the resultant monthly averaged values adequately reproduce the distribution of the observed (from GNIP stations) and modelled (by GCMs) monthly $\delta^{18}O_P$ series in northeastern Canada. In western Argentina, the distribution of monthly $\delta^{18}O_P$ values is also well reproduced but the amplitude of variation of the predicted values is too high, leading to simulated values lower than the measured ones during the colder months. The temporal $\delta^{18}O_P$ variations are positively related to air temperature given the positive coefficient $a$. In agreement with the simple Rayleigh distillation model (Dansgaard, 1964), as air temperature decreases, the specific humidity at saturation decreases, and water vapour condenses. $H_2^{18}O$ condenses preferentially, the residual water vapour gets more and more depleted as condensation proceeds. In the Tropics, the relative abundance of $^{18}O$ in the meteoric water has been observed to decrease with increasing amount of precipitation and/or relative humidity leading to a decrease in $^{18}O/^{16}O$ ratio in a high amount of precipitated water (Rozanski et al., 1993). In extra-tropical regions, $\delta^{18}O_P$ may also correlate with precipitation amount (negative coefficient $b$), since both variables depend on the meteorological conditions.

The results of the linear regressions show comparatively lower influence of precipitation on $\delta^{18}O_P$ in western Argentina than in northeastern Canada (Table 1). This suggests that the imprint of the precipitation amount on $\delta^{18}O_P$ in western Argentina is low and that $\delta^{18}O_P$ variations are mainly controlled by seasonal changes in temperature, which is in agreement with (Rozanski et al., 1995). However, due to the strong west-to-east precipitation gradient in this region (orographic rain shadow), large $\delta^{18}O_P$ variations occur over short distances (Rozanski et al., 1995; Smith and Evans, 2007; Stern and Blisniuk, 2002). Therefore, the daily precipitation dataset extracted from the gridded reanalysis data, which has a low spatial resolution (>200 km), may not represent the daily variations in precipitation at a local scale faithfully. Therefore, the model may underestimate the contribution of precipitation on $\delta^{18}O_P$ variability in this particular area.

Contrastingly, in northeastern Canada, both temperature and precipitation amount equally control the $\delta^{18}O_P$ variations. The high amount of precipitation falling in summer (~46%) should have a strong effect and decrease the $\delta^{18}O_P$ values in the condensed water, while high temperatures





counteract this effect by increasing this ratio. Before reaching northeastern Canada, the air
masses pushed by the dominant westerly winds discharge most of their humidity over the land,
leading to a depleted $\delta^{18}O_P$ signal at our sites (for the same reason, $\delta^{18}O_{TR}$ values at L20, which is
located 110 km North-East of L01, are ~1‰ lower). Interestingly, the $\delta^{18}O_P$ signal in
northeastern Canada is comparatively more depleted than in western Argentina. It is worth noting
that the resolution of the gridded meteorological dataset used here is relatively high (~10 km),
which means that the local processes are likely well represented.

**4.2. Relative performance in modelling $\delta^{18}O_{TR}$ values**
The simulated $\delta^{18}O_{TR}$ series based on daily $\delta^{18}O_P$ estimation from the regression models
reproduce the observations better than the ones based on $\delta^{18}O_P$ values derived from GCMs
(Figure 4). This is in part due to the greater number of parameters to optimize, as the calibration
process can more easily find a solution that fits the observations better. This may however reflect
error compensations especially in western Argentina where the estimated annual variability of
$\delta^{18}O_P$ is too large. Conversely in northeastern Canada, the annual variations of $\delta^{18}O_P$ that are
estimated, simulated by GCMs and observed are in good agreement (Figure SM2). Although
isotope-enabled atmospheric global models reproduce reasonably well the global distribution of
the mean annual isotope contents of the modern precipitation and their seasonality (Risi et al.,
2010), results at specific sites, especially in mountainous regions such as at our western
Argentinian site, can be less accurate (Figure SM2; see the offset between GNIP and LMDZ-
NCEP20). Ideally, daily $\delta^{18}O_P$ long-term records from meteorological stations in the study region
should be used as an input of MAIDENiso. Simulations from high-resolution regional circulation
models, such as REMOiso which has a 0.5°×0.5° (~55 km) horizontal resolution (Insel et al.,
2013; Sturm et al., 2007, 2005), may produce reliable local $\delta^{18}O_P$ values. Such dataset has proven
to be quite helpful with MAIDENiso in the Fontainebleau forest (France) (Danis et al., 2012).
However, up to now, measured or REMOiso $\delta^{18}O_P$ datasets in our regions of study do not exist,
which is the case for most regions of the world. Therefore, we recommend that daily GNIP
sations are set up in various forested ecosystems and that high resolution simulations of $\delta^{18}O_P$ are
performed in wider regions.



The modelling of $\delta^{18}O_{TR}$ values based on the estimation of $\delta^{18}O_P$ is relatively more accurate for
northeastern Canada than for western Argentina (Figure 3). As the mean levels of the measured
$\delta^{18}O_{TR}$ values are high at the western Argentinian sites (mean value of about 30‰), the Bayesian
optimization tends to increase the biochemical ($\varepsilon_0$) and kinetic ($\varepsilon_k$) fractionations as well as the
coefficient $a$, while reducing the dampening factor ($f_o$) to reach more representative mean levels
of the $\delta^{18}O_{TR}$ simulation. But still, these levels are too low in comparison with the observations
(about 2.5‰ lower; Figure SM4). When the posterior value of a calibrated parameter is limited to
the upper bound of the prior range of plausible values, as it is the case at the western Argentinian
sites for $a$, $b$ and $\varepsilon_0$ (Figure SM3), it means that either the prior range is too narrow, or the model
is inadequate, or some important process is not considered in the model. Here, the estimation of
the prior ranges of both coefficients $a$ and $b$ were based on observed (GNIP stations) and
simulated (GCMs) $\delta^{18}O_P$ values. Therefore, we expect their respective ranges to be consistent
with local processes. When the prior range of $a$ is extended to higher values in the optimization
process, observed and simulated $\delta^{18}O_{TR}$ mean levels in western Argentina are better matching.
However, in this case, the distribution of $\delta^{18}O_P$ values is shifted toward higher values, advocating
for unrealistic estimated $\delta^{18}O_P$ variations.
One other possibility is that the prior range of $\varepsilon_0$ is too narrow. In accordance with DeNiro and
Epstein (1981), Sternberg (1989) and Yakir and DeNiro (1990), the biochemical fractionation $\varepsilon_0$
is assumed here to be lower than 30‰. However, a recent study has demonstrated that this
parameter, nearly constant between 20 to 30°C, increases at lower temperatures to values of 31‰
(Sternberg and Ellsworth, 2011). During the growing season, maximum temperatures can reach
20°C in western Argentina and 30°C in northeastern Canada, which suggests that the high mean
$\delta^{18}O_{TR}$ levels in $N.$ $pumilio$ may be due to biochemical fractionation higher than 30‰ due to
temperature generally lower than 20°C. However, when the prior range of $\varepsilon_0$ is extended to 31‰
in the optimization process, the mean $\delta^{18}O_{TR}$ levels of $N.$ $pumilio$ are still too low in comparison
with the observations. These results advocate for the existence of other processes, which can
explain this offset in mean levels in Argentina. For example, higher soil water evaporation than
modelled by MAIDENiso should lead to less negative $\delta^{18}O_{SW}$ (and therefore $\delta^{18}O_{XW}$), which
could explain the high mean levels of $\delta^{18}O_{TR}$ in Argentina. Caution should be exercised with such
an interpretation since other species living in similar conditions as $N.$ $pumilio$ in western
Argentina show comparatively lower mean $\delta^{18}O_{TR}$ levels than $N.$ $pumilio$ (i.e., $Fitzroya$





*cupressoides*; see Lavergne et al. (2016)). The ongoing monitoring and evaluation of isotopic
processes based on synchronous measurements of vapour, precipitation, soil water and xylem
water will certainly help understanding the high mean levels observed in Argentina, and
increasing the representation of the involved processes in MAIDENiso.

The better fit between observed and simulated $\delta^{18}O_{TR}$ values obtained with specific forms of
synthetic distributions of daily GPP for northeastern Canada and western Argentina (Figure 3)
suggests differential limiting factors in the two regions. The synthetic bimodal distribution of
daily GPP with maxima in spring and autumn, as simulated in western Argentina, is often
observed in a diversity of ecosystems such as in the Mediterranean environments (Baldocchi et
al., 2010; Gea-Izquierdo et al., 2015). After the activation of the photosynthesis in early spring,
increasing temperatures tend to be optimal for tree growth. However, in a modelling study,
Lavergne et al. (2015) have shown that the influence of temperature on *N. pumilio*'s growth
becomes negative once a temperature threshold (soil moisture) is exceeded. Therefore, we
assume that after reaching a threshold of temperature and soil moisture summer conditions, tree
growth is not favoured, leading to a decrease of primary productivity. However, when
temperature starts to decline and soil water supply tends to increase with increasing precipitation
events, tree growth increases again until the end of the growing season. In contrast, because
precipitation is more abundant in summer (June to September) in northeastern Canada (Naulier et
al., 2014), high summer temperatures should be always beneficial to tree-growth if enough soil
water is available. Therefore, in agreement with GPP-derived eddy covariance data from the
Fluxnet network (see Gennaretti et al. (2017a)), a better fit between observations and simulations
is observed when using a unimodal rather than a bimodal GPP distribution. Monitoring of tree
physiology, environmental conditions and wood cell formation will provide a more detailed
representation of the complex biological and ecological processes operating in Patagonia,
allowing us to run the MAIDENiso model with better constraints.

**4.3. What is the main origin of the temperature signal recorded in $\delta^{18}O_{TR}$?**
The investigation of the relative contributions of the isotopic composition of the source (xylem)
water and of the $^{18}O$ enrichment of the leaf water by transpiration on the simulated $\delta^{18}O_{TR}$ reveals
that the variability of the former has a weaker influence on $\delta^{18}O_{TR}$ variations than that of the



latter in North and South America. Therefore, the temperature signal recorded in $\delta^{18}O_{TR}$ series
more likely reflects the effect of temperature on isotopic enrichment of the leaf water rather than
on the isotopic composition of the source water. At the leaf-level, air temperature has a strong
effect on the relative humidity and therefore on the vapour pressure deficit (VPD), i.e. the
difference between the saturation vapour pressure and the actual vapour pressure, which
modulates the transpiration (Barbour, 2007). Thus, the imprint of the ambient air temperature on
the fractionation processes occurring during transpiration is preferentially recorded in the tree-
rings of the two species. Furthermore, both the isotopic signature of the xylem water and of the
fractionation processes occurring at the evaporation sites of the leaves have comparatively higher
influence on $\delta^{18}O_{TR}$ in *P. mariana* than in *N. pumilio*. This is probably due to the lower amplitude
of the day-by-day variations of the relative humidity in western Argentina (SD = 5%) versus in
northeastern Canada (SD = 16%) that translates into a weaker influence of $h_{air}$ variations and
therefore of leaf-level isotopic fractionation processes on $\delta^{18}O_{TR}$ values in western Argentina
than in northeastern Canada. These results highlight the potential of MAIDENiso model to better
refine the origin of the climatic signal recorded in the oxygen isotopic signature in the tree-rings
of different species.

**5. CONCLUSION**
Here, by using MAIDENiso model, we provided a mechanistic overview of the climatic and
biological processes controlling oxygen isotopic fractionation in two American temperature-
sensitive tree species. Firstly, we have shown that using regression-based rather than model-
based $\delta^{18}O_P$ estimates as inputs increases the predictive skills of our simulations, although this
may be at the price of error compensations. Secondly, our study reveals that the variability of the
isotopic composition of the source (xylem) water has a weaker influence on $\delta^{18}O_{TR}$ variations
than that of the $^{18}O$ enrichment of the leaf water by transpiration. Finally, these findings suggest
that the imprint of temperature recorded in $\delta^{18}O_{TR}$ of the two species is likely related to the effect
of temperature on isotopic enrichment of the leaf water. The isotopic monitoring of water within
the soil-vegetation-atmosphere compartments in future work will certainly provide the input and
control data necessary to better constrain MAIDENiso. Our study demonstrates that the eco-
physiological modelling of $\delta^{18}O_{TR}$ values is necessary and likely the only approach to accurately
interpret the recorded climate signal. Based on the calibrations of MAIDENiso presented here,



the next step involves inverse modelling approaches to perform paleoclimatic reconstructions in
North and South America that are less biased by the complex and nonlinear interactions between
climate, $CO_2$ concentrations and tree growth as recommended by Boucher et al. (2014).

**ACKNOWLEDGMENTS**
A.L. has been supported by a Research associate/Lecturer position at the Aix-Marseille
University (France). F.G. has received funding from the European Union's Horizon 2020
research and innovation program under the Marie Sklodowska-Curie grant agreement No
656896. We acknowledge all data providers: the Instituto Argentino de Nivología, Glaciología y
Ciencias Ambientales (IANIGLA, Argentina) for providing the daily temperature data from La
Almohadilla site; the National Meteorological Service from Argentina for providing the monthly
temperature data from Bariloche meteorological station (Argentina); the Department of Natural
Resources Canada for providing the daily climatic data used for Quebec; the US Department of
Energy, Office of Science Biological and Environmental Research (BER) and the National
Oceanic and Atmospheric Administration Climate Program Office for providing the daily
climatic data used for Argentina; and the SWING project for providing the daily $\delta^{18}O_P$ data from
MUGCM model.

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




**Tables and Figures**

**Table 1** Definition of sensitive parameters. The posterior medians and 90% confidence intervals
are also shown.

| Parameter | Definition | Unit | Parameter type (prior range) | Values with 90% posterior confidence intervals |
|---|---|---|---|---|
| $f_o$ | Dampening factor | NA | Calibrated (0.3 to 0.5) | 0.36 [0.31; 0.46] (Arg.)<br>0.41 [0.32; 0.48] (Q.) |
| $\varepsilon_0$ | Biochemical fractionation | ‰ | Calibrated (24 to 30) | 29.99 [29.93; 30] (Arg.)<br>26.81 [24.74; 28.04] (Q.) |
| $\varepsilon_k$ | Kinetic fractionation | ‰ | Calibrated (10 to 30) | 28.86 [18.25; 29.96] (Arg.)<br>17.20 [11.16; 26.34] (Q.) |
| $a$ | Temperature dependence of $\delta^{18}O_P$ | NA | Calibrated (0.2 to 0.5 for Arg. and 0 to 0.38 for Q.) | 0.50 [0.49; 0.50] (Arg.)<br>0.31 [0.25; 0.37] (Q.) |
| $b$ | Precipitation dependence of $\delta^{18}O_P$ | NA | Calibrated (-0.3 to 0 for Arg. and -0.39 to 0 for Q.) | -0.009 [-0.15; 0] (Arg.)<br>-0.22 [-0.35; -0.14] (Q.) |
| $c$ | Intercept of $\delta^{18}O_P$ | ‰ | Fixed | -10.0 (Arg.)<br>-11.9 (Q.) |

**Table 2** Climate input data for all tested simulations

| | Daily Tmin and Tmax | Daily P | Daily $\delta^{18}O_P$ | $CO_2$ |
|---|---|---|---|---|
| **Configuration 1** | Canadian database/ NOAA-CIRES dataset | | Linear regression | Mauna |
| **Configuration 2** | Canadian database / NOAA-CIRES dataset | | MUGCM data | Loa |
| **Configuration 3** | LMDZ-NCEP20 data | | | station |




**Figure 1** Tree-ring $\delta^{18}$O time series (‰) at the three sites in Argentina (NUB, ALM and CHA in
dark grey) and two sites in Quebec (L01 and L20 in dark grey; single trees in light grey). The
bold black lines are the averaged values. The mean inter-site correlation coefficients are
r = 0.60, p < 0.05 and r = 0.80, p < 0.01 in the South and North American sites, respectively.

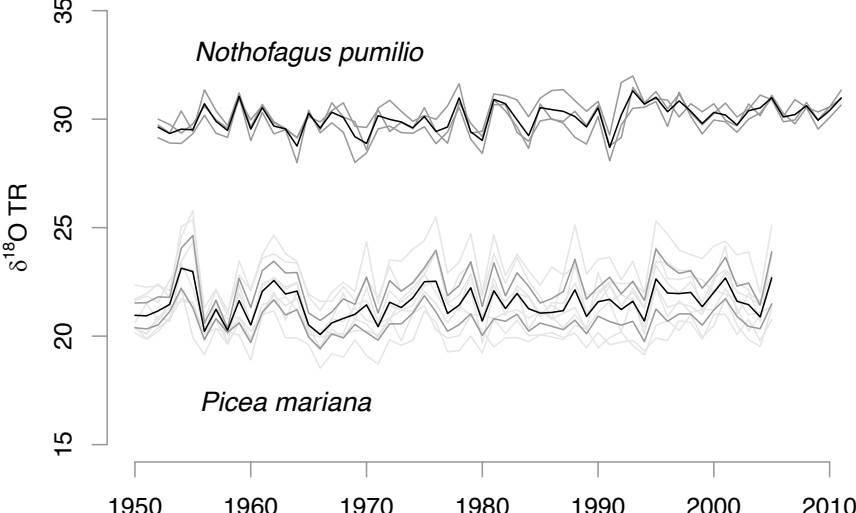




**Figure 2** Dependence of the correlation coefficients between observed and simulated $\delta^{18}O_{TR}$
series (panels A), and of the mean simulated $\delta^{18}O_{TR}$ levels (‰) (panels B) as a function of the
range of calibrated parameters $a$, $b$, $f_o$, $\varepsilon_0$ and $\varepsilon_k$ for the 50 simulations performed. In black are the
tests with the sites from Quebec and in red the oneswith the Argentinean sites. The vertical lines
are the values of a plausible block of parameters retained in the MCMC optimization. The
horizontal dashed lines are their respective 90% confidence interval calculated with 50
simulations (see Table 1). The horizontal dot lines in panel B are the mean values of the observed
$\delta^{18}O_{TR}$.

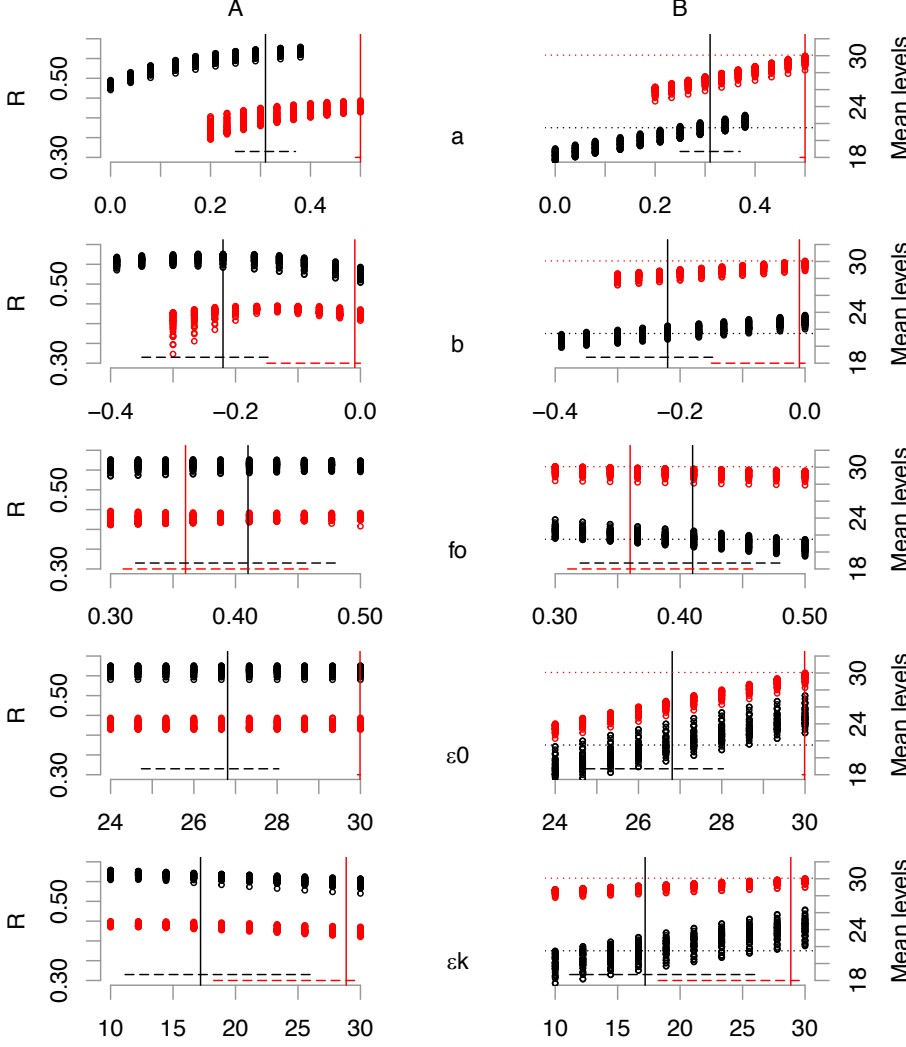




**Figure 3** Comparison between observed (red or green) and simulated (grey) $\delta^{18}O_{TR}$ chronologies in Quebec (A and B) and Argentina (C and D), respectively, using GPP (in gC.m$^{-2}$.day$^{-1}$) simulated by MAIDENiso (A and C) or synthetized for maximizing correlations (B and D). The simulations are based on estimated $\delta^{18}O_P$ series. The 50 different simulations inferred from the Markov Chain Monte Carlo (MCMC) chains are in dark grey. The ± 1 root mean square error (RMSE) range is represented in light grey. The mean correlation coefficients are significant at 99% level (**).

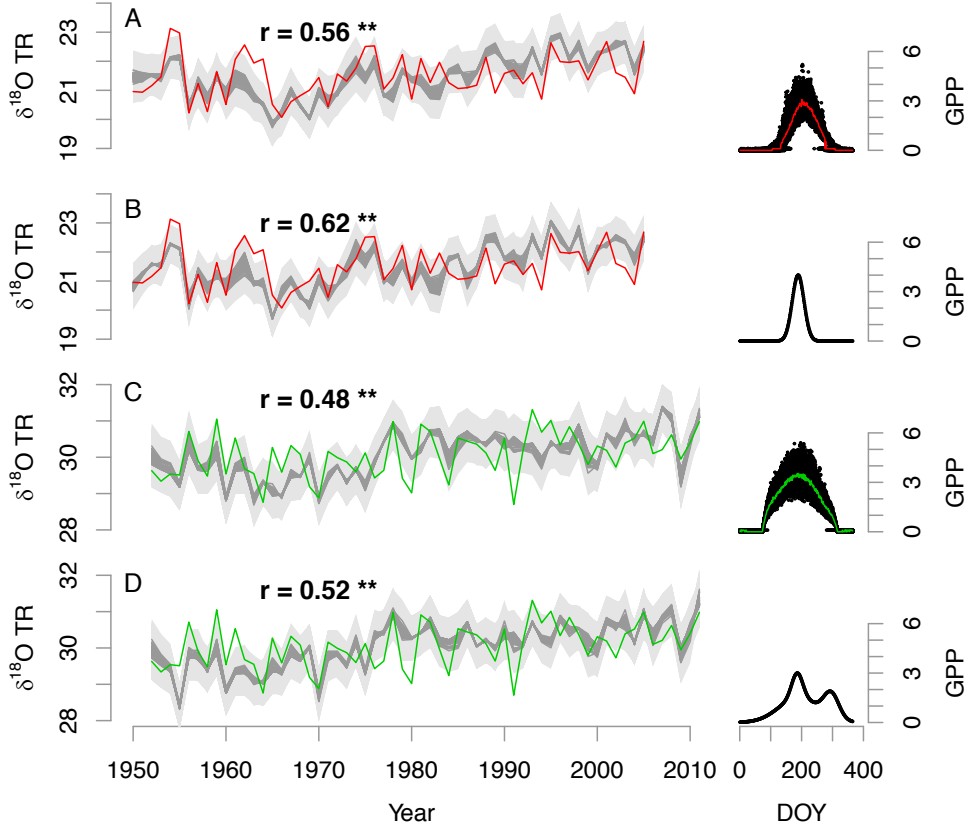



**Figure 4** Comparison of the densities of probability of the coefficient of correlation (R) between observed and simulated $\delta^{18}O_{TR}$ chronologies in Quebec and Argentina when the simulations are based on $\delta^{18}O_P$ series estimated by the regression model or from the MUGCM and LMDZ-NCEP20 models.

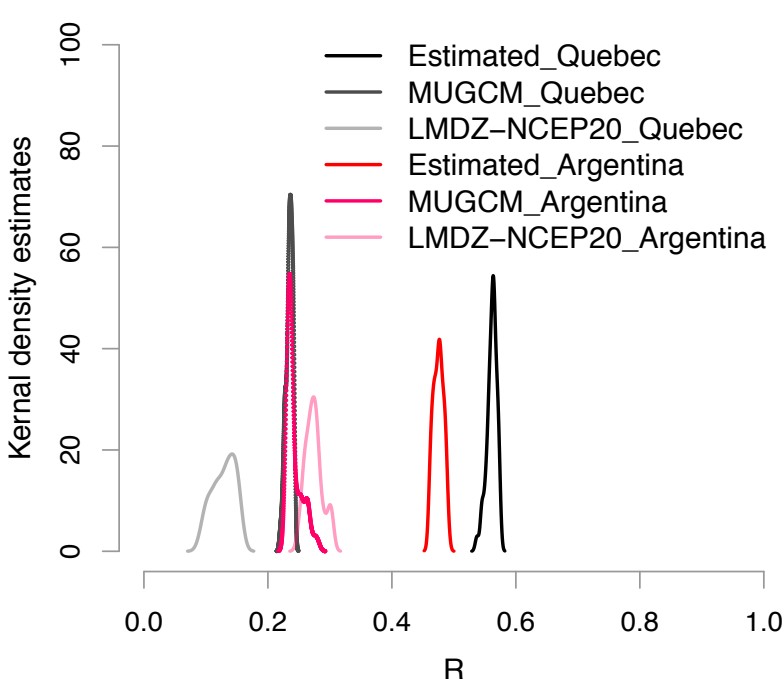



**Figure 5** Density distributions of the coefficients of determination ($R^2$) between the reference
simulations and the: 1) XW_source experiment simulation ($\delta^{18}O_V$ and $h_{air}$ set as constant, black)
and, 2) Leaf_water_enrichment_driven experiment simulation ($\delta^{18}O_{XW}$ set as constant, green) in
Quebec (bold line) and Argentina (dashed line).

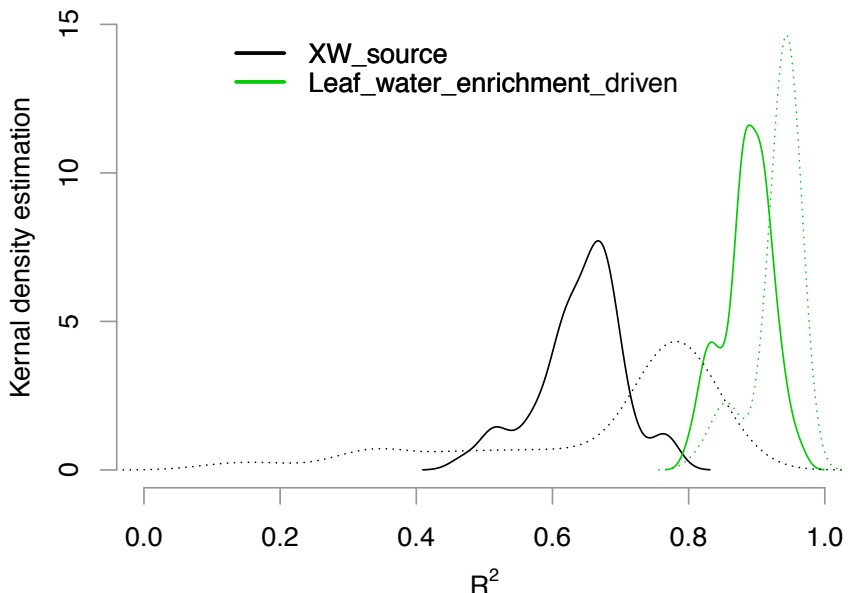


