# Peer review of "Modelling tree ring cellulose $\delta^{18}$ O variations of two temperature-sensitive tree"

_Climate of the Past, 2017_

## Referee Comment (RC1) · Anonymous Referee #1 · 18 Sep 2017

The authors test the MAIDENiso model in regard to O-isotope fractionation with "temperature-sensitive" tree species in Quebec and from Patagonia (which I interpret as ring growth being sensitive to temperature). In the case of the Canadian site, the high latitude indicates temperature sensitivity, whereas for the Argentina site the elevation probably contributes more to the temperature sensitivity. A number of parameters in the mechanistic models must be estimated, among which the estimated $\delta$18O of precipitation may have the greatest uncertainty, but parameters are also tested for sensitivity in simulating the observed tree-ring $\delta$18O. The authors found that xylem water $\delta$18O is less influential than leaf evaporative enrichment in predicting tree-ring $\delta$18O. Furthermore, temperature effects are more related to effect on leaf evaporative enrichment that T effects on precipitation isotopes.

The analysis is important and results reasonable, although there are some large $\delta$18O differences in the actual tree-ring composition between the N. American and S. American sites.

p. 35, 'tree rings' p. 101-102, 'which is an angiosperm deciduous species dominating' p. 111-112, 'In western Argentina, precipitation is largely concentrated from late fall to early spring followed by a drier and mild period during summer and early fall'... isn't late fall to early spring summer in Argentina, and therefore the following 'mile period' would be during the Argentina winter and early spring? p. 188, 'for N. pumilio, and therefore the' 216-217, 'we also used modelled daily data from the GCMs described above for both the western Argentinian and northeastern Canadian sites' 221-222, 'For the years 1950-1957,' 240 (and 159), the authors refer to 'dampening factor fo', but Eqn 1 suggests it is actually the fraction of the tree-ring $\delta$18O signal that derives from xylem water... perhaps they are synonymous? 287, in "temperature and precipitation dependences", the authors seem to mean "temperature and precipitation coefficients", i.e., a and b. 289, "more strongly" 319, what is the "reference one"? perhaps "reference simulations"? 320, what is the "source one"? perhaps "than are the XW_source simulations"? 325, what does "these results are limited upstream" mean? 341-342, change "ratio in a high amount of precipitated water" to "ratio increased higher precipitation" 362-363, why is it 'interesting(ly)' that "the $\delta$18OP signal in northeastern Canada is comparatively more depleted than in western Argentina". Given the latitude of northeastern Canada, I would expect $\delta$18OP to be isotopically lighter. 363, "northeast" 385-386, "GNIP stations" 434-435, "tree growth is inhibited, leading to a decrease of" 465, "tree rings" 719, are the "mean simulated $\delta$18OTR levels" (here in caption and in B y-axis labels) actually "$\delta$18OTR values"? or "$\delta$18OTR output"

REFERENCES The "13"s and "18"s in isotope designations in titles need to be superscripted. DeNiro and Epstein 1979, Rozanski et al. 1993, Yakir and Deniro references: too many words in title begin with upper-case letters

Figure 4, shouldn't the label on the y-axis be "kernel density"?

[Figure]

---

## Referee Comment (RC2) · Anonymous Referee #2 · 29 Sep 2017

This paper is a welcome addition to the literature on tree ring isotopes and their potential to enrich palaeoclimate reconstructions. Application of the MAIDENiso (MI) model to two different species in two different environments was undertaken, and both of the target species have the potential to provide longer palaeoclimate reconstructions. The main aims are made clear from the outset: to evaluate if MI can simulate d18O of tree rings, to identify physical processes that control d18O of tree rings using mechanistic modeling, and assess the origin of how temperature is recorded in both target species. The mixture of settings and hemispheres is also nice to see. I was also delighted by the fact that this is a well-written paper, and I enjoyed reading it.

[Figure]

I have only a few main comments that I believe can help, and a handful of minor ones. For Section 2.4. Estimation of parameters, I believe this is one of the more important elements of the study. It is my opinion here because in some cases, a range of unknowns need to be assumed or tested in a hierarchical way where observations are sparse. It might be good to mention other studies to the readers that have grappled with this issue in this section. For example, a range of unknown parameters for a Southern Hemisphere species with dendroclimatic potential was recently examined using a mechanistic model that augmented Barbour, Roden, Farquhar and Ehleringer (BRFE04). The ranges of some unknown parameters were tested simultaneously against a mean d18O chronology while others were empirically derived (Lorrey et al., 2016). The code for the model described in that paper can be found here: https://github.com/nicolasfauchereau/model_isotope

I can appreciate that some elements of MI will be different from other mechanistic models that have come before, so my pointing to the aforementioned resource is not to state it is better (or to get it cited), but rather suggesting that a myriad of modelling approaches can be helpful for distilling and probing important issues for isotope dendroclimatology.

It would also be really nice if a diagram that shows how the MI model was constructed (the main componentry and inputs, for example) could be included either in the main paper or the supplement.

Minor comments. 118-120. Reword this please as: The chronologies that were built for each species were significantly correlated between stands (Figure 1). This supported the construction of a combined isotope chronology for both the northeastern Canada and western Argentina sites. 124. please provide reference for MAIDENiso again here. If you can please provide links to the code for this model, it would be appreciated. 162. can you please cite any IAEA studies where the closest measurements would be, or have a look at whether anything useful can be gleaned from the data underpinning the online isotopes in precipitation calculator 164. First. Not Firstly. Prettification of

words by adding 'ly' is not correct grammar. 175. As above with secondly. Second. 180. can you please spell out the acronym for LMDZ5A, and also fully spell out National Centers for Environmental Protection (NCEP), as well as fully refer to the 20th Century Reanalysis (20CR) 202. I see 20CRv2c mentioned here; it should be fine, but please explain why this reanalysis dataset is chosen over something like NCEP1 or ERA-Interim. 250. Lorrey et al. (2016) evaluated the outcomes of iterative changes to unknown parameters for a d18O model output in a similar way for NZ kauri (mentioned above). This appears to be a standard way to evaluate how well a mechanistic model does for d18Otr, in a simple way. I would just mention here a range of studies that may have undertaken a similar approach to show it is an acceptable method for evaluation. 318. Leaf water enrichment (are underscores needed?) 324. Last sentence. Can you please expand on this statement just a little bit more, for clarity? 348. '….agreement with previous work (Rozanski et al)' 356. Reword to start "In contrast, in northeastern Canada…". 362. Reword to start "Of interest, the …" 374. Reword to say "Although isotope-enabled atmospheric global models can reproduce the mean annual precipitation isotopic values and seasonality for many areas (Risi et al)…" 385. Also mention here that the IAEA datasets that had a good deal of chemistry run on them in the 1970-80s may have been compromised by pan evaporation and therefore enrichment. Have to treat many of those extant (older) data sources very carefully. 471. Firstly. As above. 473. Secondly. As above. 475. Last instead of Finally. References. Some errors with author names (Farquhar was one) please check this carefully.

Lorrey, A.M., Brookman, T., Evans, M.N., Fauchereau, N.C., Barbour, M.M, Macinnis-Ng, C., Criscitiello, A.S., Eischeid, G., Horton, T.W., Fowler, A.M., Schrag, D.P. 2016. Stable oxygen isotope signatures of early season wood in New Zealand kauri (Agathis australis) tree rings: Prospects for palaeoclimate reconstruction.ÂăDendrochronologia,Âă40, 50-63.Âă

---

## Author Comment (AC1) · 3 Oct 2017

We thanks a lot Reviewer#1 for all his comments and suggestions. We are responding to each of his comments below as Author Comment (AC).

Comments p. 35, 'tree rings'

AC: We changed this part as proposed by Reviewer#1 (L30).

p. 101-102, 'which is an angiosperm deciduous species dominating'

AC: We changed this part as proposed by Reviewer#1 (L103).

[Figure]

p. 111-112, 'In western Argentina, precipitation is largely concentrated from late fall to early spring followed by a drier and mild period during summer and early fall': isn't late fall to early spring summer in Argentina, and therefore the following 'mile period' would be during the Argentina winter and early spring?

AC: In western Argentina, precipitation is concentrated in late fall to early spring (May-November) followed by a drier and mild period during summer and early fall (December-April). We specified the respective months for each period in the text (L112-114).

p. 188, 'for N. pumilio, and therefore the'

AC: We changed this part as proposed by Reviewer#1 (L199).

216-217, 'we also used modelled daily data from the GCMs described above for both the western Argentinian and northeastern Canadian sites'

AC: We changed this part as proposed by Reviewer#1 (L230-231).

221-222, 'For the years 1950-1957,'

AC: We changed this part as proposed by Reviewer#1 (L235).

240 (and 159), the authors refer to 'dampening factor fo', but Eqn1 suggests it is actually the fraction of the tree-ring $\delta$18O signal that derives from xylem water: perhaps they are synonymous?

AC: The 'dampening factor' is defined in the literature (e.g. Saurer et al. 1997) as the proportion of oxygen atoms that is exchanged between sucrose and xylem water during cellulose synthesis. It is modelled as a coefficient in Eqn1 to take into account the part of $\delta$18O signal derived from xylem water during this exchange that is incorporated in the cellulose $\delta$18O.

287, in "temperature and precipitation dependences", the authors seem to mean "temperature and precipitation coefficients", i.e., a and b.

AC: Yes, the temperature and precipitation dependences are modelled as coefficients a and b, respectively.

289, "more strongly"

AC: We changed this part as proposed by Reviewer#1 (L307).

319, what is the "reference one"? perhaps "reference simulations"?

AC: Yes, it is the reference simulations. We changed it in the text (L342).

320, what is the "source one"? perhaps "than are the XW_source simulations"?

AC: Idem, we changed it in the text (L343).

325, what does "these results are limited upstream" mean?

AC: We removed this sentence in the text that was not clear (L352).

341-342, change "ratio in a high amount of precipitated water" to "ratio increased higher precipitation"

AC: We have simplified the sentence to be more understandable: 'Consequently, in the Tropics, the 18O/16O ratio in the meteoric water has been observed to decrease with increasing amount of precipitation and/or relative humidity.' (L365-367).

362-363, why is it 'interesting(ly)' that "the $\delta$18Op signal in northeastern Canada is comparatively more depleted than in western Argentina". Given the latitude of northeastern Canada, I would expect $\delta$18Op to be isotopically lighter.

AC: We expanded a little bit the explanation of why the $\delta$18Op signal in northeastern Canada was comparatively more depleted than in western Argentina, followingReviewer#1 recommendations (L397-399).

363, "northeast"

AC: We decided to keep 'northeastern', which is often used (L394).

385-386, "GNIP stations" AC: We changed this part as proposed by Reviewer#1 (L414).

434-435, "tree growth is inhibited, leading to a decrease of"

AC: We changed this part as proposed by Reviewer#1 (L484).

465, "tree rings"

AC: We changed this part as proposed by Reviewer#1 (L507-508).

719, are the "mean simulated $\delta$18OTR levels" (here in caption and in B y-axis labels) actually "$\delta$18OTR values"? or "$\delta$18OTR output"

AC: Yes, they are the simulated $\delta$18OTR values. We decided to stay with 'simulated $\delta$18OTR levels' because in this figure we want to show that some parameters are affecting the mean levels of $\delta$18OTR values (L890).

REFERENCES The "13"s and "18"s in isotope designations in titles need to be super-scripted. DeNiro and Epstein 1979, Rozanski et al. 1993, Yakir and Deniro references: too many words in title begin with upper-case letters

AC: We have corrected all the errors detected by Reviewer#1 in the reference list.

Figure 4, shouldn't the label on the y-axis be "kernel density"?

AC: We think that the y-axis as 'kernel density estimates' is fine (L908).

―――――――――――――――――

---

## Author Comment (AC2) · 3 Oct 2017

We are happy that the reviewer really liked and enjoyed our study. We are responding to his comments and suggestions as Author Comment (AC).

Comments: I have only a few main comments that I believe can help, and a handful of minor ones. For Section 2.4. Estimation of parameters, I believe this is one of the more important elements of the study. It is my opinion here because in some cases, a range of unknowns need to be assumed or tested in a hierarchical way where observations are sparse. It might be good to mention other studies to the readers that have grappled with this issue in this section. For example, a range of unknown

parameters for a Southern Hemisphere species with dendroclimatic potential was recently examined using a mechanistic model that augmented Barbour, Roden, Farquhar and Ehleringer (BRFE04). The ranges of some unknown parameters were tested simultaneously against a mean $\delta$18O chronology while others were empirically derived (Lorrey et al., 2016). The code for the model described in that paper can be found here: https://github.com/nicolasfauchereau/model_isotope

AC: We have added in the text the link to the code of MAIDENiso model: https://doi.org/10.6084/m9.figshare.5446435.v1 (L134-135).

I can appreciate that some elements of MI will be different from other mechanistic models that have come before, so my pointing to the aforementioned resource is not to state it is better (or to get it cited), but rather suggesting that a myriad of modelling approaches can be helpful for distilling and probing important issues for isotope dendroclimatology.

AC: We thank Reviewer#2 for this suggestion. We have mentioned in the revised manuscript other studies that have used this approach: e.g. Danis et al., 2012; Lorrey et al., 2016 (see L266-268).

It would also be really nice if a diagram that shows how the MI model was constructed (the main componentry and inputs, for example) could be included either in the main paper or the supplement.

AC: Different publications have already detailed the construction of the MAIDENmodel (among the most recent one, Danis et al., 2012 and Gennaretti et al. 2017b). We have cited these papers in the text as references (L126 and L128-129).

Minor comments. 118-120. Reword this please as: The chronologies that were built for each species were significantly correlated between stands (Figure 1). This supported the construction of a combined isotope chronology for both the northeastern Canada and western Argentina sites.

AC: We changed this part as proposed by Reviewer#2 (L119-122).

124. please provide reference for MAIDENiso again here. If you can please provide links to the code for this model, it would be appreciated.

AC: We have added the references on MAIDENiso model mentioned above (L126 and L128-129).

162.can you please cite any IAEA studies where the closest measurements would be, or have a look at whether anything useful can be gleaned from the data under pinning the online isotopes in precipitation calculator

AC: To our knowledge, no IAEA studies have been developed in the regions of our study. In Argentina, only studies further north (30°S; Rozanski et al. 1995) and further south (47°-48°S; Stern and Blisniuk, 2002) have been done to understand the variability of $\delta18Op$. We are referring to the IAEA dataset in the text (L171-172) and we discussed the studies that have been done further north and further south in the Discussion section (L371-376).

164. First. Not Firstly. Prettification of words by adding 'ly' is not correct grammar.

AC: We have changed it as proposed by Reviewer#2 (L174).

175. As above with secondly. Second.

AC: We have changed it as proposed by Reviewer#2 (L185).

180. can you please spell out the acronym for LMDZ5A, and also fully spell out National Centers for Environmental Protection (NCEP), as well as fully refer to the 20th Century Reanalysis (20CR)

AC: LMDZ5A is the acronym of 'Laboratoire de Météorologie Dynamique Zoom'. We have spelled out all the acronyms in the text as proposed by Reviewer#2 (L190-192).

202. I see 20CRv2c mentioned here; it should be fine, but please explain why this

reanalysis dataset is chosen over something like NCEP1 or ERA-Interim.

AC: We have used the 20CRv2c dataset to extract daily minimum-maximum temperatures and precipitation amount because it is one of the few reanalysis products covering entirely the 20th century. Furthermore, NCEP1 has been replaced by 20CRv2c and ERA-Interim starts in 1979. We add this explanation in the text (L218-219).

250. Lorrey et al. (2016) evaluated the outcomes of iterative changes to unknown parameters for a $\delta$18O model output in a similar way for NZ kauri (mentioned above). This appears to be a standard way to evaluate how well a mechanistic model does for $\delta$18OTR, in a simple way. I would just mention here a range of studies that may have undertaken a similar approach to show it is an acceptable method for evaluation.

AC: As suggested by Reviewer#2, we have added other studies that have undertaken a similar approach (L266-268).

318. Leaf water enrichment (are underscores needed?)

AC: We have deleted the underscores (L280, L287 and L341).

324. Last sentence. Can you please expand on this statement just a little bit more, for clarity?

AC: A suggested as well by Reviewer#1, we have deleted this last sentence, which was not clear (L352).

348. '. . .agreement with previous work (Rozanski et al)'

AC: We changed it as proposed by Reviewer#2 (L373).

356. Reword to start "In contrast, in northeastern Canada. . .".

AC: We changed it as proposed by Reviewer#2 (L391).

362. Reword to start "Of interest, the . . ."

AC: We changed this sentence as proposed by Reviewer#1 (L397-399).

374. Reword to say "Although isotope-enabled atmospheric global models can reproduce the mean annual precipitation isotopic values and seasonality for many areas (Risi et al)..."

AC: We changed the sentence as proposed by Reviewer#2 (L410-412).

385. Also mention here that the IAEA datasets that had a good deal of chemistry run on them in the 1970-80s may have been compromised by pan evaporation and therefore enrichment. Have to treat many of those extant (older) data sources very carefully.

AC: We have incorporated this explanation as well in the text (L421-436).

471. Firstly. As above.

AC: We changed it as proposed by Reviewer#2 (L521).

473. Secondly. As above.

AC: We changed it as proposed by Reviewer#2 (L523).

475. Last instead of Finally.

AC: We changed it as proposed by Reviewer#2 (L525).

References. Some errors with author names (Farquhar was one) please check this carefully.

AC: As already mentioned in the response to Reviewer#1, we have corrected all the errors detected in the Reference list.

---

## Author Response (AR1)

**Point-by-point response to the reviews**

Responses to the reviewers are shown in red as Author Comment (AC). The number of the lines where to find the modifications done in the text are those from the Track changes version of the manuscript (see below).

**Anonymous Referee #1**

The authors test the MAIDENiso model in regard to O-isotope fractionation with "temperature-sensitive" tree species in Quebec and from Patagonia (which I interpret as ring growth being sensitive to temperature). In the case of the Canadian site, the high latitude indicates temperature sensitivity, whereas for the Argentina site the elevation probably contributes more to the temperature sensitivity. A number of parameters in the mechanistic models must be estimated, among which the estimated $\delta^{18}O$ of precipitation may have the greatest uncertainty, but parameters are also tested for sensitivity in simulating the observed tree-ring $\delta^{18}O$. The authors found that xylem water $\delta^{18}O$ is less influential than leaf evaporative enrichment in predicting tree-ring $\delta^{18}O$. Furthermore, temperature effects are more related to effect on leaf evaporative enrichment that T effects on precipitation isotopes. The analysis is important and results reasonable, although there are some large $\delta^{18}O$ differences in the actual tree-ring composition between the N. American and S. American sites.

AC: We thanks a lot Reviewer#1 for all his comments and suggestions.

Comments
p. 35, 'tree rings'

AC: We changed this part as proposed by Reviewer#1 (L30).

p. 101-102, 'which is an angiosperm deciduous species dominating'

AC: We changed this part as proposed by Reviewer#1 (L103).

p. 111-112, 'In western Argentina, precipitation is largely concentrated from late fall to early spring followed by a drier and mild period during summer and early fall': isn't late fall to early spring summer in Argentina, and therefore the following 'mile period' would be during the Argentina winter and early spring?

AC: In western Argentina, precipitation is concentrated in late fall to early spring (May-November) followed by a drier and mild period during summer and early fall (December-April). We specified the respective months for each period in the text (L112-114).

p. 188, 'for N. pumilio, and therefore the'

AC: We changed this part as proposed by Reviewer#1 (L199).

216-217, 'we also used modelled daily data from the GCMs described above for both the western Argentinian and northeastern Canadian sites'

AC: We changed this part as proposed by Reviewer#1 (L230-231).

221-222, 'For the years 1950-1957,'

AC: We changed this part as proposed by Reviewer#1 (L235).

(and 159), the authors refer to 'dampening factor fo', but Eqn1 suggests it is actually the fraction of the tree-ring $\delta^{18}O$ signal that derives from xylem water: perhaps they are synonymous?

AC: The 'dampening factor' is defined in the literature (e.g. Saurer et al. 1997) as the proportion of oxygen atoms that is exchanged between sucrose and xylem water during cellulose synthesis. It is modelled as a coefficient in Eqn1 to take into account the part of $\delta^{18}O$ signal derived from xylem water during this exchange that is incorporated in the cellulose $\delta^{18}O$.

287, in "temperature and precipitation dependences", the authors seem to mean "temperature and precipitation coefficients", i.e., a and b.

AC: Yes, the temperature and precipitation dependences are modelled as coefficients a and b, respectively.

289, "more strongly"

AC: We changed this part as proposed by Reviewer#1 (L307).

319, what is the "reference one"? perhaps "reference simulations"?

AC: Yes, it is the reference simulations. We changed it in the text (L342).

320, what is the "source one"? perhaps "than are the XW_source simulations"?

AC: Idem, we changed it in the text (L343).

325, what does "these results are limited upstream" mean?

AC: We removed this sentence in the text that was not clear (L352).

341-342, change "ratio in a high amount of precipitated water" to "ratio increased higher precipitation"

AC: We have simplified the sentence to be more understandable: 'Consequently, in the Tropics, the $^{18}O/^{16}O$ ratio in the meteoric water has been observed to decrease with increasing amount of precipitation and/or relative humidity.' (L365-367).

362-363, why is it 'interesting(ly)' that "the $\delta^{18}O_p$ signal in northeastern Canada is comparatively more depleted than in western Argentina". Given the latitude of northeastern Canada, I would expect $\delta^{18}O_p$ to be isotopically lighter.

AC: We expanded a little bit the explanation of why the $\delta^{18}O_p$ signal in northeastern Canada was comparatively more depleted than in western Argentina, followingReviewer#1 recommendations (L397-399).

363, "northeast"

AC: We decided to keep 'northeastern', which is often used (L394).

385-386, "GNIP stations"
AC: We changed this part as proposed by Reviewer#1 (L414).

434-435, "tree growth is inhibited, leading to a decrease of"

AC: We changed this part as proposed by Reviewer#1 (L484).

465, "tree rings"

AC: We changed this part as proposed by Reviewer#1 (L507-508).

719, are the "mean simulated $\delta^{18}O_{TR}$ levels" (here in caption and in B y-axis labels) actually "$\delta^{18}O_{TR}$ values"? or "$\delta^{18}O_{TR}$ output"

AC: Yes, they are the simulated $\delta^{18}O_{TR}$ values. We decided to stay with 'simulated $\delta^{18}O_{TR}$ levels' because in this figure we want to show that some parameters are affecting the mean levels of $\delta^{18}O_{TR}$ values (L890).

REFERENCES
The "13"s and "18"s in isotope designations in titles need to be superscripted. DeNiro and Epstein 1979, Rozanski et al. 1993, Yakir and Deniro references: too many words in title begin with upper-case letters

AC: We have corrected all the errors detected by Reviewer#1 in the reference list.

Figure 4, shouldn't the label on the y-axis be "kernel density"?

AC: We think that the y-axis as 'kernel density estimates' is fine (L908).

**Anonymous Referee #2**

This paper is a welcome addition to the literature on tree ring isotopes and their potential to enrich palaeoclimate reconstructions. Application of the MAIDENiso (MI) model to two different species in two different environments was undertaken, and both of the target species have the potential to provide longer palaeoclimate reconstructions. The main aims are made clear from the outset: to evaluate if MI can simulate $\delta^{18}O$ of treerings, to identify physical processes that control $\delta^{18}O$ of tree rings using mechanistic modeling, and assess the origin of how temperature is recorded in both target species.

The mixture of settings and hemispheres is also nice to see. I was also delighted bythe fact that this is a well-written paper, and I enjoyed reading it.

AC: We are happy that the reviewer really liked and enjoyed our study.

I have only a few main comments that I believe can help, and a handful of minorones. For Section 2.4. Estimation of parameters, I believe this is one of the moreimportant elements of the study. It is my opinion here because in some cases, arange of unknowns need to be assumed or tested in a hierarchical way where observationsare sparse. It might be good to mention other studies to the readers that have grappled with this issue in this section. For example, a range of unknown parametersfor a Southern Hemisphere species with dendroclimatic potential was recently examined using a mechanistic model that augmented Barbour, Roden, Farquhar and Ehleringer (BRFE04). The ranges of some unknown parameters were tested simultaneouslyagainst a mean $\delta^{18}O$ chronology while others were empirically derived (Lorrey et al., 2016). The code for the model described in that paper can be found here: https://github.com/nicolasfauchereau/model_isotope

AC: We have added in the text the link to the code of MAIDENiso model: https://doi.org/10.6084/m9.figshare.5446435.v1 (L134-135).

I can appreciate that some elements of MI will be different from other mechanisticmodels that have come before, so my pointing to the aforementioned resource is notto state it is better (or to get it cited), but rather suggesting that a myriad of modelling approaches can be helpful for distilling and probing important issues for isotope dendroclimatology.

AC: We thank Reviewer#2 for this suggestion. We have mentioned in the revised manuscript other studies that have used this approach: e.g. Danis et al., 2012; Lorrey et al., 2016 (see L266-268).

It would also be really nice if a diagram that shows how the MI model was constructed (the main componentry and inputs, for example) could be included either in the mainpaper or the supplement.

AC: Different publications have already detailed the construction of the MAIDENmodel (among the most recent one, Danis et al., 2012 and Gennaretti et al. 2017b). We have cited these papers in the text as references (L126 and L128-129).

Minor comments.
118-120. Reword this please as: The chronologies that were built foreach species were significantly correlated between stands (Figure 1). This supported the construction of a combined isotope chronology for both the northeastern Canada and western Argentina sites.

AC: We changed this part as proposed by Reviewer#2 (L119-122).

124. please provide reference for MAIDENiso again here. If you can please provide links to the code for this model, it would be appreciated.

AC: We have added the references on MAIDENiso model mentioned above (L126 and L128-129).

162.can you please cite any IAEA studies where the closest measurements would be, orhave a look at whether anything useful can be gleaned from the data underpinningthe online isotopes in precipitation calculator

AC: To our knowledge, no IAEA studies have been developed in the regions of our study. InArgentina, only studies further north (30°S; Rozanski et al. 1995) and further south (47°-48°S; Stern and Blisniuk, 2002) have been done to understand the variability of $\delta^{18}O_p$. We are referring to the IAEA dataset in the text (L171-172) and we discussed the studies that have been done further north and further south in the Discussion section (L371-376).

164. First. Not Firstly. Prettification of words by adding 'ly' is not correct grammar.

AC: We have changed it as proposed by Reviewer#2 (L174).

175. As above with secondly. Second.

AC: We have changed it as proposed by Reviewer#2 (L185).

180. can you please spell out the acronym for LMDZ5A, and also fully spell out National Centers for Environmental Protection (NCEP), as well as fully refer to the 20th Century Reanalysis (20CR)

AC: LMDZ5A is the acronym of 'Laboratoire de Météorologie Dynamique Zoom'. We have spelled out all the acronyms in the text as proposed by Reviewer#2 (L190-192).

202. I see 20CRv2c mentioned here; it should be fine, butplease explain why this reanalysis dataset is chosen over something like NCEP1 orERA-Interim.

AC: We have used the 20CRv2c dataset to extract daily minimum-maximum temperatures and precipitation amount because it is one of the few reanalysis products covering entirely the 20th century. Furthermore, NCEP1 has been replaced by 20CRv2c and ERA-Interim starts in 1979. We add this explanation in the text (L218-219).

250. Lorrey et al. (2016) evaluated the outcomes of iterative changes tounknown parameters for a $\delta^{18}O$ model output in a similar way for NZ kauri (mentioned above). This appears to be a standard way to evaluate how well a mechanistic modeldoes for $\delta^{18}O_{TR}$, in a simple way. I would just mention here a range of studies that mayhave undertaken a similar approach to show it is an acceptable method for evaluation.

AC: As suggested by Reviewer#2, we have added other studies that have undertaken a similar approach (L266-268).

318. Leaf water enrichment (are underscores needed?)

AC: We have deleted the underscores (L280, L287 and L341).

324. Last sentence. Can youplease expand on this statement just a little bit more, for clarity?

AC: A suggested as well by Reviewer#1, we have deleted this last sentence, which was not clear (L352).

348. '…agreement with previous work (Rozanski et al)'

AC: We changed it as proposed by Reviewer#2 (L373).

356. Reword to start "In contrast, in northeastern Canada…".

AC: We changed it as proposed by Reviewer#2 (L391).

362. Reword to start "Of interest, the …"

AC: We changed this sentence as proposed by Reviewer#1 (L397-399).

374. Reword to say "Although isotope-enabled atmospheric global models can reproduce the mean annual precipitationisotopic values and seasonality for many areas (Risi et al)…"

AC: We changed the sentence as proposed by Reviewer#2 (L410-412).

385. Also mention here that the IAEA datasets that had a good deal of chemistry run on them in the 1970-80s may have been compromised by pan evaporation and therefore enrichment. Haveto treat many of those extant (older) data sources very carefully.

AC: We have incorporated this explanation as well in the text (L421-436).

471. Firstly. As above.

AC: We changed it as proposed by Reviewer#2 (L521).

473. Secondly. As above.

AC: We changed it as proposed by Reviewer#2 (L523).

475. Last instead of Finally.

AC: We changed it as proposed by Reviewer#2 (L525).

References. Some errors with author names (Farquhar was one) please check this carefully.

AC: As already mentioned in the response to Reviewer#1, we have corrected all the errors detected in the Reference list.

**List of relevant changes made in the manuscript**

- Following the reviewer's suggestions, we have rewritten some sentences in the manuscript (see track changes' version of the manuscript).
- We have added a link to the code of the model (L134-135): https://doi.org/10.6084/m9.figshare.5446435.v1

- We have extended our discussion on the GNIP datasets quality (L421-436).

[revised manuscript text omitted]

Aliénor Lavergne 2/10/y 15:58

Aliénor Lavergne 2/10/y 15:58

Aliénor Lavergne 2/10/y 15:58

Aliénor Lavergne 2/10/y 15:58

---

## Author Response (AR2)

**Response to editor comments**

Comments to the Author:

Dear Dr. Lavergne,

Thank you for submitting a revised version of your manuscript "Modelling tree-ring cellulose d18O variations of two temperature-sensitive tree species from North and South America". Please find a few additional minor suggestions and typos to correct. I also invite you to have another careful read of your manuscript, including figure legends and references.

Best regards,

Laurie Menviel

AC : We thanks the Editor for her additional suggestions in the text.

L. 44: "is more likely linked to the" do you want to say "is primarily linked to"?

AC : Yes. We have changed this part of the sentence as suggested by the Editor.

Move L. 133-134 (location of the model) to a new section "Code and data availability" to be added at the end of manuscript (right before the acknowledgment section.

Move L. 165-166 (IAEA web address) to the new section "Code and data availability"

Move L. 181-182: "extracted at http://paos.colorado.edu/~dcn/SWING/database.php" to the new section "Code and data availability"

Move L. 209: "provided by the NOAA/OAR/ESRL (https://www.esrl.noaa.gov/psd/data/gridded/data.20thC_ReanV2c.html)," to the new section "Code and data availability"

Move L. 220: " http://cfs.nrcan.gc.ca/projects/3/4" to the new section "Code and data availability"

AC : We have added a 'Code and data availability' section before the Acknowledgement one as suggested by the Editor. We have moved in this new section all the references to each datasets used.

Please modify as follow:

L. 180: "1) The Melbourne University model (MUGCM)"

L. 183: "the Laboratoire de Météorologie Dynamique Zoom model (LMDZ5A)"

Please rephrase: L. 227 "as displayed in the observations" or "as shown"

L. 324: Please rephrase "are more highly correlated" as I don't think it is grammatically correct. Some suggestions: "are highly correlated" or "have a higher correlation coefficient"

Reviewer 1 suggested to use "tree rings" please use this spelling throughout the manuscript (i.e. change all the "tree-rings", e.g. title, section 2.1, legend of figure 1, keywords…).

L. 476: "we have provided"

L. 733: please add a space between "ones" and "with" or rephrase: "The tests sites from Quebec are in black and the Argentinean ones are in red."

Legend of figure 3: Please spell out "DOY"

AC : We have changed all the parts mentioned above as suggested by the Editor.

Legend of figure 4: "densities of probability of the coefficient of correlation" should it be instead "probability density functions"?

AC : We have changed 'densities of probability' into 'density distributions' in the legend of Figure 4, as done in Figure 5.

Figure 4 and 5: As already noted by Reviewer 1 the y axis should be "Kernel" and not "Kernal", please correct. Also in figure 5 the y axis should read "Kernel density estimate"

AC : We have corrected 'Kernal' into 'Kernel' in Figures 4 and 5, and the y axis legend of Figures 5 and SM3.

[revised manuscript text omitted]

Aliénor Lavergne 9/10/y 14:14

Aliénor Lavergne 9/10/y 14:14

Aliénor Lavergne 9/10/y 14:14

Aliénor Lavergne 9/10/y 14:14

Aliénor Lavergne 9/10/y 14:14

Aliénor Lavergne 9/10/y 14:14

Aliénor Lavergne 9/10/y 14:14

Aliénor Lavergne 9/10/y 14:14

**Figure 3** Comparison between observed (red or green) and simulated (grey) $\delta^{18}O_{TR}$ chronologies
in Quebec (A and B) and Argentina (C and D), respectively, using GPP (in $gC.m^{-2}.day^{-1}$)
simulated by MAIDENiso for each day of the year (DOY) (A and C) or synthetized for
maximizing correlations (B and D). The simulations are based on estimated $\delta^{18}O_P$ series. The 50
different simulations inferred from the Markov Chain Monte Carlo (MCMC) chains are in dark
grey. The ± 1 root mean square error (RMSE) range is represented in light grey. The mean
correlation coefficients are significant at 99% level (**).

[Figure]

**Figure 4** Density distributions of the coefficient of correlation (R) between observed and simulated $\delta^{18}O_{TR}$ chronologies in Quebec and Argentina when the simulations are based on $\delta^{18}O_P$

series estimated by the regression model or from the MUGCM and LMDZ-NCEP20 models.

Aliénor Lavergne 9/10/y 14:14

Aliénor Lavergne 9/10/y 14:14

[Figure]

**Figure 5** Density distributions of the coefficients of determination ($R^2$) between the reference
simulations and the: 1) XW source experiment simulation ($\delta^{18}O_V$ and $h_{air}$ set as constant, black)
and, 2) Leaf water enrichment driven experiment simulation ($\delta^{18}O_{XW}$ set as constant, green) in
Quebec (bold line) and Argentina (dashed line).

[Figure]

[Figure]

Aliénor Lavergne 9/10/y 14:14